# Experience of depression in older adults with and without a physical long-term condition: findings from a qualitative interview study

Lydia Poole ,[1] Rachael Frost ,[2] Hannah Rowlands,[3] Georgia Black [4]

[1]Institute of Health Informatics, UCL, London, UK
[2]Research Department of Primary Care and Population Health, University College London, London, UK
[3]Institute of Psychiatry, Psychology & Neuroscience, King's College London, London, UK
[4]Applied Health Research, University College London, London, UK

**Correspondence to**
Dr Lydia Poole;
lydia.poole@ucl.ac.uk

## ABSTRACT

**Objective** To understand how the lived experience of depression differs among patients with a long-term condition (LTC) compared with those without an LTC, and how the experience differs across different types of LTC.

**Design** Face-to-face, semistructured interviews.

**Setting** Primary care; General Practitioner (GP) surgeries in and around North London.

**Participants** 41 primary care patients with depression were recruited. Our sample comprised participants aged 55–75 years with depression only (n=12), depression and coronary heart disease (n=5), depression and type 2 diabetes (n=10) and depression and arthritis (n=14).

**Results** Interviews were conducted, audio recorded, transcribed and analysed using thematic analysis. The results revealed that the cardinal diagnostic symptoms of depression (anhedonia, sadness) were experienced by all our participants regardless of LTC. However, the LTC did interact with depression by compounding somatic, cognitive and emotional symptoms, increasing disability and reducing independence, and hindering attempts at coping with mental illness. Our findings demonstrate common experiences across patients as well as key differences based on LTC.

**Conclusions** We suggest four key implications for future care practices of these patients: (1) not all participants with depression and LTC view their mental and physical health as interconnected; there should be allowances in care plans for separate treatment pathways; (2) key features of depression that affect LTC management are social withdrawal and lack of motivation to self-manage or access healthcare; (3) key features of LTCs that worsen depression are pain, the unpredictability of future health and progressive disability; (4) positive self-management of LTC could improve self-efficacy and therefore mood, and should be encouraged.

## Strengths and limitations of this study

► Participant diagnoses derived from clinical records and confirmed by self-report.
► Participants were drawn from across multiple sites across North London, with diversity in demography achieved.
► Comparative thematic analysis allowing for comparisons within and between groups.
► Small sample of depression and coronary heart disease cohort.
► Majority female in the depression and arthritis cohort.

## INTRODUCTION

Estimates from the UK Biobank study suggest the prevalence for a single episode of major depressive disorder is 6.4%, varying to 12.2% for moderate recurrent major depression and 7.2% for severe recurrent major depression,[1] among British adults. Depression has been associated with increased all-cause mortality[2]; however, causality has not yet been determined.[3] One possible explanation is that individuals with depression are more than twice as likely to have a concomitant physical long-term condition (LTC), such as diabetes and coronary heart disease (CHD).

It has been hypothesised from quantitative investigations of epidemiological and clinical cohort studies that depressive symptoms which occur in cardiac patients are distinct from those experienced in psychiatric populations based on three key differences: composition of symptoms, timeline and treatment responsiveness.[4] With regard to symptoms, the hallmarks of psychiatric depression are persistent sadness and loss of interest and pleasure in activities (anhedonia). These are often associated with cognitive changes such as a sense of worthlessness and guilt, among others. By contrast, there is growing recognition that depressive symptoms in people with physical illness are dominated by somatic symptoms, such as fatigue and loss of energy.[5 6] In addition, it has been suggested that the chronicity, and the timing of onset of depressive symptoms in relation to the LTC, have implications for morbidity and mortality.[7 8] Finally, it appears that different treatment strategies may be appropriate for those living with psychiatric

depression compared with those who have comorbid depression,[9] with particular benefit coming from integrative collaborative care models of healthcare.[10] Qualitative data have revealed that depression in physical illness has been shown to have been normalised by healthcare practitioners leading to barriers in its management[11] and there may be resistance to antidepressant medication among some LTC patients.[12] Given these empirical observations surrounding the differences in depression symptoms, timeline and treatment, some authors have suggested that a separate subtype of depression may exist, driven by biological (inflammatory) changes[13] which in turn are implicated in LTCs.[14] However, despite a plausible hypothesis for the existence of a separate subtype of depression in those with an LTC, whether or not this translates to differences in patient experience is unclear. Therefore, further qualitative research is warranted to examine whether the lived experience of depression is similar or different in patients with depression and/or a comorbid LTC.

Furthermore, experiences of depression across LTCs are likely to differ. A metasynthesis of qualitative data has suggested that patients with depression and a comorbid LTC either see their mental and physical health as either separate and independent from each other, or interrelated with one having caused or worsened the other.[15] However, the similarities and/or differences in how depression is experienced in different LTCs are currently not well understood. For example, depression in diabetes has been revealed to be heterogenous with aspects of diabetes-related distress and demoralisation being apparent,[16] while depression in CHD has been associated with loss, both in terms of interpersonal loss as well as loss of health/control.[12]

Therefore, the aim of this study was twofold. First, we aimed to examine how the lived experience of depression is similar or different between those with an LTC and those without, and second, we aimed to understand how experiences of depression are similar or different between types of LTC. In order to take into account the correlation between age and LTC onset, we focused on the experiences of depression in adults in mid-late life.

## METHODS
### Participant selection and recruitment
Thirty-seven participants were recruited using mail-outs from 14 UK general practices in the Barnet and Enfield regions of North London; a further one participant was recruited using opportunistic sampling from General Practitioner (GP) surgeries in North West London; and a final three participants were recruited from the Camden and Islington's Improving Access to Psychological Therapy (IAPT) service.

Our inclusion criteria led us to recruit participants who identified as male and female, aged 55–75 years and English speaking. All participants were screened for current depression by participating GP practices via electronic and hand searches of medical records. Current depression was defined as a patient who had either received a clinical code of depression by their GP and/or had received a referral or prescription for depression treatment in the preceding 2 months. All participants were medically stable outpatients living within the community. We excluded patients with a severe cognitive, mental or psychological impairment that would impede their ability to provide informed consent. Moreover, to compare patients living with depression, with and without an LTC, we recruited patients to one of four participant groups according to additional diagnoses identified in their medical notes: depression only, depression and CHD, depression and type 2 diabetes (T2D) and depression and arthritis. To reduce the cross-over between the groups, participants with more than one LTC of interest (ie, diabetes and arthritis) in their medical notes were excluded. We also excluded those with another LTC such as cancer, stroke and chronic lung disease to allow us to focus our discussions more easily on our chosen LTCs. The LTCs were selected based on a priori knowledge to elicit a variety of illness experiences: CHD can have an acute, life-threatening presentation (eg, acute coronary syndromes); T2D involves monitoring and self-management of blood sugar levels and can lead to acute episodes of illness (hypoglycaemia); arthritis is associated with pain and functional impairment.

Practice GPs reviewed the list of those to approach prior to mail-out and removed people who met the exclusion criteria or they considered inappropriate to contact. The mail-out included full details of the study and interested patients were invited to contact the research team to discuss participation. Patients were screened by researchers over the telephone using the same inclusion and exclusion criteria before arranging a convenient time and place for the interview to take place. We used purposive sampling to capture a range of clinical and demographic characteristics within each group where possible; specifically, among those patients who expressed an interest in taking part, we prioritised telephone screening to allow us to recruit a diverse sample by sex, age, ethnicity and location.

### Interview data collection
We conducted 1:1 semistructured, face-to-face interviews mainly at participants' homes or in a meeting room at University College London (UCL) (or another convenient location if requested), lasting approximately 1 hour (range 42–97 min). Interviews were carried out between May and October 2019. Data collection continued within this time frame and ceased once all eligible patients at participating sites had been contacted. Interviews were conducted by two female researchers: HR (MSc; research assistant) (n=40) or LP (PhD; senior research fellow) (n=1). The interviewers had received training in qualitative interviewing, and HR had prior experience of conducting qualitative interviews with vulnerable groups. Neither HR or LP had any prior connection with

participants, other than the contact made during the recruitment process where the study objectives and procedures were explained. The interviews began by presenting participants with four short vignettes for participants to read and reflect on. The vignettes contained the experiences of fictional characters written in the third person (see online supplemental appendix 1) and were designed to encourage participants to reflect on their own symptoms of depression. This method is well established for eliciting detailed information on sensitive topics in qualitative research.[17] The interview then covered a number of topics to explore the experience of living with depression: diagnosis, precipitating events, symptoms, change in experience over time, management of depression and treatment preferences. In participants with a comorbid LTC, we also explored the impact of depression on their physical illness and vice versa. The topic guides (see online supplemental appendix 2) were modified as interviews progressed through team discussions. To circumvent issues surrounding stigma, initial questions steered clear of clinical labels, and instead invited participants to reflect on 'changes to their mood'. As each interview progressed, the interviewer used the participants' own terminology. After each interview, demographic data and self-reported medical information were collected. The primary interviewer (HR) kept a reflective diary in order to document thoughts, feelings and other interesting contextual information that occurred to her. Participants were provided with relevant local and national mental health service information and a £10 retail voucher honorarium. Any concerns over participant safety (including suicidal ideation and self-harm) were discussed with the senior researcher (LP) before contacting participants' GP with their consent. A standard operating procedure was in place for this. Interviews were audio recorded, with brief fieldnotes, transcribed verbatim by an external company and anonymised and verified for accuracy by the research team (HR and LP). Data for this study are archived in a public data repository.[18]

## Patient and public involvement

Patients and the public were not involved in the development of the research question, design or study implementation. However, the results of this study were disseminated via a large outreach and engagement event through social media and podcasting platforms.

## Data analysis

We used reflexive thematic analysis[19 20] to analyse our data from a realist/essentialist perspective in which a (mostly) one-way relationship is taken to operate between meaning, experience and language. According to this approach, data saturation is not required, rather meaning is conceived as developing from the interpretation of the data.[21] All transcripts contained the entire interview, including the discussion that took place regarding the material in the vignettes. The data arising from the discussion of vignettes varied across participants; some

participants did not engage with the vignettes and the conversation quickly flowed to the semistructured interview questions, whereas for others it prompted a divulgement of prior personal events. As such, these data were not separated, but rather the entire transcript was treated as a single data piece. Transcripts were read by HR (a health psychology researcher expertise in LTCs) and LP (a health psychology researcher with an interest in depression and multimorbidity); GB (a health services researcher with expertise in health improvement) and RF (a health services researcher with expertise in ageing and mental health) read a subsample. Data analysis was led by LP. Transcripts from each group were first read and analysed inductively, separately, to enable us to build an understanding of the idiographic qualities of the data set[19]; inductive analysis is a bottom-up approach where the codes and themes are driven by the data and not by a theoretical framework. Each set of transcripts was read repeatedly, and codes were developed by LP in conjunction with HR. HR's reflective diary was consulted to ensure no additional meaning or interpretation could be gleaned from the data. Discrepancies were discussed until consensus was reached. Coding of transcripts was performed using NVivo V.12 (QSR International, 2018). A codebook was kept to organise ideas and for note keeping. For example, our coding tree included codes for the onset and timeline of depressive symptoms, stressful life events, cognitive symptoms of depression, somatic symptoms of depression, past decisions, social connections, emotional response to illness, physical illness impacting daily living, the symptoms of the physical illness, approaching death, coping strategies and help seeking, and attitudes towards care. Codes were then grouped to construct themes; refined through team discussions and piloting. After conducting the initial analysis across the entire data set, we tabulated the themes and subthemes using Excel (Microsoft 365 for Windows 10) to allow us to systematically explore the similarities and differences in experiences across groups. This approach was informed by framework analysis[22] and has been combined with thematic analysis by previous researchers conducting comparative qualitative analysis.[23] The indexing and charting of the data enabled us to maintain rigour in the analytical process. Themes were then written out and refined with feedback from the team.

## RESULTS
### Participants

Out of 137 patients who expressed an interest in the study, 45 were eligible and 41 completed the interviews. Reasons for ineligibility included not meeting the clinical inclusion criteria (n=37), non-responsive to return phone calls (n=26), not interested (n=3), unavailable for interview (n=1), sample size deemed sufficient for the cohort (n=23) and non-English speaker (n=2). Participants were grouped into four cohorts based on their medical records and verified during screening telephone calls: depression

**Table 1** Characteristics of the sample, stratified by group

| | Whole sample (N=41) | Depression only (n=12) | Depression and CHD (n=5) | Depression and T2D (n=10) | Depression and arthritis (n=14) |
| --- | --- | --- | --- | --- | --- |
| | M±SD or N (%) | M±SD or n (%) | M±SD or n (%) | M±SD or n (%) | M±SD or n (%) |
| Sex (female) | 24 (58.5) | 6 (50.0) | 1 (20.0) | 4 (40.0) | 13 (92.9) |
| Age (years) | 63.5 (5.5) | 62.4 (5.5) | 66.0 (4.0) | 61.0 (5.2) | 65.1 (5.9) |
| Married/cohabiting | 17 (41.5) | 6 (50.0) | 2 (40.0) | 5 (50.0) | 4 (28.6) |
| Ethnicity | | | | | |
| White British | 27 (65.9) | 10 (83.3) | 2 (20.0) | 6 (60.0) | 9 (64.3) |
| Indian | 3 (7.3) | | | 2 (20.0) | 1 (7.1) |
| Pakistani | 1 (2.4) | | 1 (20.0) | | |
| Black Caribbean | 1 (2.4) | | 1 (20.0) | | |
| Other White background | 7 (17.1) | 2 (16.7) | 1 (20.0) | 1 (10.0) | 3 (21.4) |
| Other Asian background | 2 (4.9) | | | 1 (10.0) | 1 (7.0) |
| Employment status | | | | | |
| Employed full time | 9 (22.0) | 3 (25.0) | 1 (20.0) | 3 (30.0) | 2 (14.3) |
| Employed part-time | 1 (2.4) | 1 (8.3) | | | |
| Self-employed | 7 (17.1) | 3 (25.0) | | | 4 (28.6) |
| Retired | 15 (36.6) | 4 (33.3) | 3 (60.0) | 3 (30.0) | 5 (35.7) |
| Unemployed | 4 (9.8) | 1 (8.3) | | 3 (30.0) | |
| Volunteer | 2 (4.9) | | 1 (20.0) | | 1 (7.1) |
| Disabled | 3 (7.3) | | | 1 (10.0) | 2 (14.3) |
| Highest educational qualification—degree or higher | 15 (36.6) | 3 (25.0) | 3 (60.0) | 4 (40.0) | 5 (35.7) |
| Antidepressant usage (yes) | 35 (85.4) | 9 (75.0) | 4 (80.0) | 9 (90.0) | 13 (92.9) |

CHD, coronary heart disease; T2D, type 2 diabetes.

only (n=12), depression and CHD (n=5), depression and T2D (n=10) and depression and arthritis (n=14). In terms of CHD, patients in this group had received a diagnosis for myocardial infarction (n=1), angina (n=2) and heart failure (n=2). In terms of arthritis, patients in this group included those with rheumatoid arthritis (n=2) and osteoarthritis (n=12). Overall, participants were aged 63.5 years (SD 5.5) and we sampled roughly equal numbers of men and women. However, the majority of depression and arthritis participants were female (93%). Approximately one-third of the sample were from a non-White ethnic group and 85.4% of all participants were taking antidepressant medication(s) at the time of the interview. The characteristics of the sample are presented in table 1.

**Thematic overview**
We generated five overarching themes that encompass the experience of living with depression in those with and without an LTC. In each theme, we have drawn out the relationship between the experience of depression in those with and without an LTC and, where relevant, compared this across the different LTC groups (CHD, T2D and arthritis). Across all the themes, we found that the experience of depression and LTCs acted to compound the effects of each other; this was particularly apparent for those whose illness had resulted in changes to daily living. Figure 1 presents a schematic diagram of our themes.

**Theme 1: shaped by the course of time**
Across our sample, depression was not static; it was shaped by individuals' life histories and past events. The experience of depression also shifted over time so that as people's lives progressed, different triggers for new episodes became more apparent. In those with an LTC, hardship brought about by their LTC and a loss of health was described by some as a trigger for depression.

*Depression onset (first-ever episode)*
In participants with and without LTCs, the onset of the first-ever episode of depression occurred at different ages; many could identify roots for their depression in childhood or adolescence, while others had received a diagnosis more recently. Across the groups, most participants were able to identify a precipitating adverse life event(s) to their first-ever depressive episode such as financial insecurity, (multiple) bereavement, a relationship breakdown, children taken into care and physical and sexual abuse. For some, seemingly smaller grievances had accumulated over time to initiate the first-ever episode of depression.

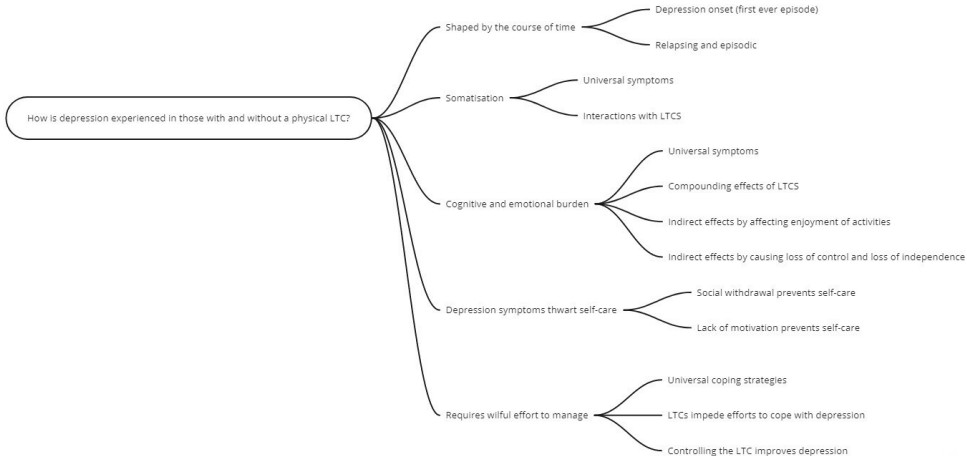

**Figure 1** Schematic diagram of theme structure. LTC, long-term condition.

In contrast to the depression-only participants, in participants with an LTC, life histories revealed two separate discourses relating to the onset of the initial depression diagnosis: coincidental and interconnected. As such, for some, the LTC was coincidental to the onset of the depression, with the depression and LTC diagnoses occurring at different times and in relation to different precipitating/surrounding events. This discourse was predominately discussed by those with T2D and, to a lesser extent, the CHD cohort. These individuals had a clear narrative about the onset of their depression, which was related to other life pressures such as caring responsibilities, fraud allegations and work rather than to their LTC:

> I don't think so. I don't really think so. I had no qualms about my health. It was the other things. The inferiority complex, the pressure of work and the unhappiness with my married life. It was those things. (P39, depression and T2D, male)

> I honestly can't say that [depression] is the reason of that [CHD]. But yes, when you have I suppose chest pain you think about it, you sit down, you [pause]. But I don't think it took me into depression at that time or anxiety or something. (P45, depression and CHD, male)

The second discourse highlighted the interconnected nature of the initial depression and LTC diagnosis. For these participants, depression was a new experience, and was directly attributed to their LTC, or its implications (eg, hospitalisations, loss of mobility, job loss). The type of LTC did not make a difference, only whether the LTC was associated with unpredictable, major and irreversible upheaval.

> Depressed? It was when I came out of hospital after having my triple bypass because my life changed completely. I lost my job, my relationship broke up, the kids and all that and I just sort of had a sort of breakdown and that was it really. (P44, depression and CHD, male)

### Relapsing and episodic
Across all participants, after the initial onset, the experience of depression had a varying time course. A few participants had new onset depression. However, for the majority of participants, their history of depression was chronic, with repeated episodes over time. This had led to a familiarity with when a new 'bout' was coming on, even if there was no discernible trigger.

Where there was a clear trigger for a new episode of depression, this was often a difficult life event, such as a child going to prison or a transition to retirement. However, for some people with chronic, pre-existing depression, the LTC had also triggered a new episode. This led to a new experience of depression for these participants, which was not centred around their earlier difficult life circumstances which had originally brought about their depression. Instead, the LTC raised fears surrounding declining health, death and frailty, which fed into the new depressive episode:

> The depression since than that was different. It was felt trapped then … I was full of tension, full of uncomfortable feeling with him. I just couldn't, it was like having a snake you know circling around you know… So that is thank God I don't have that. The depression that I feel in this and that is because of my health problems and that bothers me, that bothers me a lot. (P22, depression and CHD, female)

### Theme 2: somatisation
Somatic symptoms of depression were frequently experienced by our participants, in those with and without an LTC. The experience was largely similar across our four cohorts. Differences did exist in sleep disturbance and crying, with LTCs (mainly arthritis but also T2D) contributing to the experience of these somatic symptoms.

### Universal somatic symptoms
Somatic symptoms were discussed by all our participants, irrespective of LTC. Symptoms discussed included

appetite changes (including overeating and undereating), crying and loss of intimacy. By far, the most common of all the somatic complaints was sleep disturbance. Some described difficulties falling asleep, restless and disturbed sleep, vivid dreams and nightmares and difficulty getting out of bed in the morning. Psychomotor retardation was another common somatic complaint linked to fatigue, which was experienced across all groups, irrespective of LTC.

> Really, really very slow, very sluggish. Very physically, very physical. In fact, almost as physical as it is mental…. Yeah like you're swimming. Like you're swimming against a sea of something coming at you. And you're trying to wade yourself through and knowing that if you, and no energy, zero physical energy. (P13, depression only, female)

### Somatic symptoms and interactions with LTCs
For those with an LTC, the illness also directly contributed to the experience of somatic symptoms of depression. Sleep was negatively affected by the LTC through worries about managing the condition (as with P2 with T2D) or pain. Pain was particularly associated with participants in the depression and arthritis cohort.

> I haven't been a very good sleeper lately; I don't sleep the full 8 hours that they say you should do. I do think about things and one of the things obviously I do think about is make sure my diabetes is correct. (P2, depression and T2D, male)

> I can't sleep at night because of pain… And even at night I get a lot of pain in my back and that so I am just putting my legs round 10 times this way and 10 times that way because I can't sleep. (P4, depression and arthritis, female)

While crying was frequently described across all our participants, in our arthritis participants we observed a unique contribution of the LTC. Arthritis (and in particular arthritic pain) compounded the emotional and somatic experience of depression, and could prompt tears of fear and frustration.

> I am chopping up the onions, I have to go and sit down because the pain in my back is such that I can't, I have to sit down. And like I have cried with it, not just from the onions but real tears, through frustration I suppose and pain. (P41, depression and arthritis, female)

### Theme 3: cognitive and emotional burden
In this theme, the participants' lived experiences of the cognitive and emotional symptoms of depression are described. These key diagnostic symptoms of depression were clearly experienced by all our participants. However, we observed compounding and indirect routes linking LTCs with the cognitive and emotional burden

of depression; moreover, thoughts of death were particularly prominent within the depression and CHD group.

### Universal cognitive and emotional symptoms
Participants, regardless of LTC, had an overwhelming sense of disinterest in the world around them; they described a deep-rooted sadness, hopelessness, worthlessness and low self-esteem. Some felt as though they were being punished.

> You get what you deserve in life. And I don't deserve nothing. You know, I wasn't a good father when I was young. I was selfish. And it's come back and bit me up the bum, bit me on the bum…. (P12, depression only, male)

### Compounding effects of LTC on cognitive and emotional depressive symptoms
For some, the combination of the LTC and the depression led to a compounding effect on the cognitive and emotional depressive symptoms. In this way, the LTC directly contributed to the participants' experience of depression. This was particularly observed in those with T2D and arthritis.

> I'm sitting there and taking the blood and giving myself injections and some days I just think, you know, how come it's come to this? And I know how it's come to this because, you know, I'm not managing my life in any sensible way, and I know that. (P25, depression and T2D, male)

For participants with severe arthritic pain, pain was seen as an all-consuming and debilitating experience, which led to misery.

> I'm getting pain back in my right knee and I'm going to have to eventually have that one done. It's really, oh god, what does that feel like? When I'm not feeling … oh … it feels like having been almost injected with negativity. (P28, depression and arthritis, female)

For P32, her depression and the functional limitations imposed by her arthritis were difficult to unravel, with both acting in synergy to lead to a sense of hopelessness:

> I'm also physically unable to do as much as I would like to do in terms of just - housework and shopping and it's difficult to pull it apart which is leading the sort of feelings of hopelessness and you know is it worth it and I can't, I might as well just stay in bed. (P32, depression and arthritis, female)

### Indirect effects of LTC on cognitive and emotional depressive symptoms through affecting enjoyment of activities
For some of our LTC participants, the depression was a greater burden than their LTC, so cognitive and emotional depressive symptoms were experienced regardless of their LTC. In this way, some of our LTC participants experienced feelings of anhedonia, worthlessness, guilt,

sadness, etc, similarly to the depression-only participants. However, for others, the LTC (notably T2D and arthritis) fed into the experience of cognitive and emotional symptoms by removing their enjoyment of previously pleasurable activities.

An example comes from P2, whose diabetes directly contributed to his feelings of anhedonia. This participant was particularly concerned about managing his blood glucose levels during social situations.

> You are part of it because you are invited, you all go out, you are all friends, you have known one another for years and everything is there, but you don't feel as if you are really taking part if you get what I am saying. So that's the part of the mood I have found since I have been diagnosed with diabetes and that's the thing that holds me back from properly being 100% committed to enjoying it. (P2, depression and T2D, male)

Pain also featured heavily in the descriptions of altered lifestyles recounted by our arthritis participants, perpetuating feelings of anhedonia.

> That's, and I do enjoy my food, I do enjoy cooking. I mean I haven't had a dinner party now for over a year. I have always enjoyed doing that. But that's, and it's like washing up, you know I can do so much, […] but you know like my washing up is in stages. I will do this first, then go and sit down, come back and do that lot, the next lot, then go and sit down. […]. That is where the arthritis and depression, they are like, they join forces. (P41, depression and arthritis, female)

### Indirect effects of LTC on cognitive and emotional symptoms by causing a loss of control and independence

Financial insecurity and unequal personal relationships were experienced by many of our participants, irrespective of LTC, and had contributed to feelings of being out of control. Deteriorating physical health (particularly for those with arthritis) and increasingly frailty had also led to feelings of loss of control for some. This was linked to feelings of loss of independence, which in turn led to feelings of depression.

> Some of my clothes are high up, I can never reach them unless somebody is there, that makes me depressed having to rely on somebody all the time. (P4, depression and arthritis, female)

### Thoughts of dying

Suicidal ideation, thoughts of death (but not suicide) and actual suicide attempts were disclosed in participants with and without LTCs. Although the presence of an LTC did not influence suicidal thoughts, having CHD was an important leveller for these participants. These participants lived with the unpredictable nature of their illness and the threat of sudden and irreversible loss of health:

> Even now I still think about it because when I just took a deep breath here, I felt a bit of a twinge there and it's when I got told about it, it was very disturbing because I was shown the blockage […] and although I've been reassured that it won't go up, block my heart or anything like that, but it makes you wonder because you never know what might happen. (P30, depression and CHD, male)

For P44, his brush with death due to having a heart attack has given him a greater acceptance of death:

> Well I just didn't give a shit about anything anymore, you know, apparently according to them I died twice, so I didn't give a shit no more about it, death doesn't scare me no more. (P44, depression and T2D, male)

For others with CHD, they were also having to confront death, but they discussed their fear surrounding a prolonged or sudden death:

> I keep saying to my sister, 'I don't care if I die actually in my sleep that will be fine. But if I don't die and I can't move, like I said to you if I can't move and I can't get up and can't call that horrifies me.' That is enough to, well you know. (P22, depression and CHD, female)

> So, since that day, perhaps I have, it wasn't so much when I had the heart bypass, but since the [aortic] aneurism and everything else, […] I worry that every time I go to bed, I don't know if I'm going to wake up. You know. So, I worry about that. (P30, depression and CHD, male)

### Theme 4: depression symptoms thwart self-care

In this theme, we draw on the ways in which the experience of depression impeded our participants' ability to take care of themselves; many described not washing or brushing their teeth, not wanting to cook proper meals and retreating from their social network. This was apparent for all our participants but added an extra layer of complexity to participants with an LTC.

### Depression leads to lack of motivation to self-care

Depression was associated with an apathy which led to a loss of interest in behaviours that might lead to improved self-care. This was true of behaviours that might improve the depression itself such as engaging in social activities and exercising, as well as those important for managing LTCs.

The impact of depression on attempts to control diabetes could lead to self-neglect, both in terms of medication adherence and adherence to medical appointments.

> I suppose the most obvious impact is there are days when I just can't be bothered to take my medication, both the medication for the depression and the insulin for my diabetes and I know it's stupid […]. (P25, depression and T2D, male)

There are times now when I wouldn't not turn up for an appointment, absolutely, I just […] But yes, there are times when I just think 'what's the bloody point?' in dealing with daily stuff. (P25, depression and T2D, male)

Exercise was seen as an important self-care behaviour for both mental and physical health among our participants. However, P32 described how her lack of motivation to exercise was contributing to her physical decline, which in turn compounded her feelings of depression.

As for the physical side of things, again, I suspect they would like [me] to say, I'll lose weight, get more exercise and it's very difficult when you don't want to go out of the house, particularly I haven't the confidence to go into a gym or to a swimming pool, it's not really, you know. […] So, and the other fear is, of course, the older I get, the general, you know, aging and infirmities, you know, yeah, it worries me greatly. You know, and it doesn't help the depression to lift. (P32, depression and arthritis, female)

### Social withdrawal prevents self-care behaviours

Many participants described how their attempts to cope had been thwarted by the depression itself. This was particularly true for those whose depression had led to social withdrawal. The retreat from daily life saw participants becoming withdrawn from both the outside world and their family and friends. Social withdrawal impacted on participants' management of their LTC. This was observed particularly in those with T2D and arthritis.

I did find it hard to leave the house sometimes, you know. You'd just sort of eat what was there rather than go shopping, you know. (P27, depression and T2D, female)

Well they said because of the way, it's not helped, because I stopped, I wouldn't get out of the house, I was staying in my room, that can affect arthritis because then you are letting it seize up more. (P3, depression and arthritis, female)

As seen with those experiencing a lack of motivation, social withdrawal also negatively impacted participants' willingness to engage with medical professionals (both for depression and/or the LTC).

I mean even the doctor, I have got to phone her tomorrow, they are going to arrange for the nurse to come out because I am overdue blood tests and blood pressure and that […]. Because she knows that I just don't want to leave the house again. (P3, depression and arthritis, female)

### Theme 5: requires wilful effort to manage

The experience of depression was described as an active process, requiring an individual to find ways to cope. For some with an LTC, efforts to cope with their depression

were thwarted by the functional limitation imposed by their illness. Controlling the LTC was also important to help manage the depression.

### Universal coping strategies

While many of our participants struggled to cope, nearly all engaged in some strategies to navigate their current situation such as the use of alcohol and food, medical treatments and drawing on friendships and family members for support. Often this included attempts to mask feelings as a means to find comfort; alcohol, in particular, was frequently described as a way to relax and forget.

But what I have done, and I have been guilty of, is purchasing alcohol and trying to blot it out with the alcohol. (P23, depression and arthritis, female)

Many participants were engaged with healthcare professionals for depression management. For some, antidepressants had been beneficial, making their symptoms less severe. However, for others, side effects or fears surrounding dependency prevented their use.

Most of us are given medication, which makes you zombie, anyway. It doesn't help you. It just block your pains, you know. You are numb. You are not sorting out. And how long are you going to use those medication? (P24, depression and T2D, female)

### LTCs impede efforts to cope with depression

Having an LTC prevented some participants from engaging with coping strategies to help manage their depression. For those with arthritis, this included being unable to take part in activities that promote well-being, including hobbies.

But I think, yeah, so coping, well I am not walking as much as I want to, I have stopped going, I used to go to water workout, water aerobics every week and I have stopped doing that, partly because of the difficulty of getting into my bathers and afterwards drying myself. (P1, depression and arthritis, female)

As P32 described, the combination of her depression and arthritis makes coping with each insurmountable:

You know so the depression makes you lack motivation to, perhaps, I say modify some of the effects of the arthritis, but the arthritis also, in limiting you, makes the depression seem worse. Together it seems like a huge, insurmountable problem because if, on their own, they might be manageable, […]. So it is, yes, it impedes any kind of self-care or therapy, if you like, together.

### Controlling the LTC improves depression

Interestingly, across all LTC groups, those who successfully managed to engage with strategies to cope with their illness, saw benefits to their mental health through feeling an increased sense of control/self-efficacy. For example,

failure to self-manage insulin dosages appropriately negatively impacted P25's mood, while being in control of his T2D also helped him self-manage his depression.

> But I carry glucose around with me. … I think we were away somewhere and, but I was completely useless for about two hours and I suppose I, you know, I had a moment of clarity about the need to keep on top of it properly, because I sort of knew that if I didn't, not only would I feel crap, physically, the idea of feeling crap physically was depressing. (P25, depression and T2D, male)

Interestingly, P30 with CHD found the unpredictable nature of his cardiac symptoms very distressing, but the use of his own blood pressure monitor helped him regain a sense of control.

> This is my life saver, my BP [blood pressure] machine, I do it every day, […] I was told 140 over 80, right, yes, 140, 80, so once it's within this, I go to sleep a bit better. If I didn't know what my BP was, I'd be worried even more. (P30, depression and CHD, male)

For some participants, the physical limitations imposed by their LTC led to social isolation. This was particularly apparent for those with arthritis. If participants received support with things that their arthritis prevented (eg, housework, food shopping), this was seen as positive for managing their depression, allowing them to live well within the limits of the LTC.

> So yeah, [lady friend's name] and [man friend's name] took me out yesterday to go to Aldi, it's the first time since February that I have gone out to the shops. (P3, depression and arthritis, female)

## DISCUSSION

This paper explored the lived experience of depression in primary care patients with and without a comorbid physical LTC (CHD, T2D and arthritis). Using thematic analysis, we generated five overarching themes. 'Shaped by the course of time' revealed a shifting, non-linear experience of depression in which the experience was defined by earlier life experiences as well as more minor, cumulative events. For some, the LTC diagnosis had caused catastrophic upheaval to participants' lives; for these individuals the LTC was capable of triggering depression (both first-ever or a recurrent episode). For many, other life events were more prominent to determining the course of depression; for these participants the LTC was coincidental, running alongside the depression but not triggering new episodes. 'Somatisation' highlighted the physical experience of depression in our participants; for many, these symptoms were as dominant as the cognitive and emotional symptoms. Somatic symptoms were universally experienced across all our groups, in particular feelings of tiredness and fatigue and changes to eating habits. Where LTCs did interact with the somatic symptoms, this

was mainly driven by arthritic pain contributing to and compounding these symptoms. 'Cognitive and emotional burden' explored the impact of mental symptoms, such as feelings of worthlessness, guilt, punishment, anhedonia and sadness, on participants. LTCs compounded these cognitive and emotional experiences as well as indirectly contributing to a worsening of these symptoms by affecting participants' enjoyment of previously pleasurable activities and causing a loss of control and independence. Thoughts of dying were frequently experienced, and prominent in patients with CHD where the unpredictability of future health was a psychological challenge. Theme 4, 'Depression symptoms thwart self-care', examined how depression itself acts as a barrier to engagement in self-care behaviours. Loss of motivation and social withdrawal were key aspects of the depressive experience which thwarted attempts for LTC self-management (particularly those with T2D). Finally, 'Requires wilful effort to manage' revealed universal engagement in many coping strategies, particularly attempts to mask depressive feelings such as the use of alcohol and food. However, LTCs (particularly arthritis) did impede participants' ability to cope with depression, while successful management of the LTC had a positive impact on mood.

Findings from quantitative studies continue to make attempts to understand the relationship between depression in LTCs, with the causal model thought to be bidirectional.[9] Our findings supported the notion that the LTC was capable of triggering depression; however, this was restricted to those whose LTC has caused major life upheaval (eg, loss of employment, relationship breakdown). It has been estimated that around 80% of initial depressive episodes occur after a stressful life event[24]; and that the severity of stressors required to precipitate recurrent depressive episodes decreases over time.[25] None of our participants described their depression acting as a precursor to their LTC. In fact, for many, the LTC was experienced as coincidental to the onset of depression; this is a view that has been supported by a previous qualitative metasynthesis.[15]

The difference in the experience of depression in those with and without an LTC has not been studied previously using qualitative research. Quantitative research has identified differences in terms of symptom presentation, chronicity and timing of symptoms and treatment responsivity.[4] Our findings have added to this by revealing that while the symptoms of depression (somatic, cognitive, affective) were universally experienced by those with and without depression, the LTC contributed to the burden of these symptoms by either directly provoking them (eg, arthritic pain leading to tears of frustration) or by indirectly leading to negative mood by undermining the enjoyment of partaking in previously pleasurable activities as well as reducing individuals' sense of independence. T2D and arthritis were key in this regard since both illnesses were associated with lifestyle changes which negatively impacted the participants' depression, increasing feelings of anhedonia, sadness, guilt and worthlessness. Previous

research has explored the impact of arthritis on leisure activities, finding that functional and physical constraints imposed by the illness can affect self-esteem.[26] Our findings extend this to the experience of those with arthritis *and* depression, suggesting that these limitations worsen the symptoms of depression. Participants with diabetes also saw a compounding effect of their physical condition on their mental health. This was particularly true for those who described being distressed by T2D and its management. Indeed, diabetes distress is now recognised as a separate disorder to major depression[27] which has been linked to feelings of burden, difficulties adhering to medication regimes and depression.[28]

Loss of control and lack of independence featured heavily in the accounts of our participants with arthritis; whereby functional limitations imposed by the LTC indirectly impacted depression by removing individual's sense of control and autonomy. Loss was a key theme identified in the qualitative UPBEAT-UK study of patients with depression and CHD, where patients described loss of health and interpersonal loss.[12] Health deterioration featured heavily in the accounts of all our participants with a physical LTC, and while thoughts of dying were universally experienced across groups, our CHD group revealed a strong sense of an unpredictable and uncontrollable death, which was not apparent in those with T2D and arthritis. This realistic fear underpinned the experience of depression in these participants.

Our findings also observed differences in the experience of depression in participants' ability to self-care and to cope with their mental health. Social withdrawal and lack of motivation were commonly experienced in all our participants. We described how these features of depression thwart attempts to engage with self-care behaviours such as personal hygiene, meal preparation, etc, as well as LTC (particularly T2D) specific illness management behaviours such as adhering to medication and attending hospital appointments. While depression has been associated with poor medication adherence in previous quantitative studies of patients with T2D and CHD,[29 30] there has been a lack of qualitative studies describing the hows and whys of this phenomenon; more research in this area is warranted.

Findings from our study revealed the adoption of common coping behaviours across all our participants such as using alcohol and food to mask symptoms and drawing on expert sources of support; this supports findings from earlier work.[12] However, importantly, we observed that the presence of an LTC interacted with participants' attempts to manage their depression. Those with an LTC, and in particular those with increasing disability from their arthritis, were prevented from engaging with coping mechanisms they had previously adopted to promote mental well-being. On the upside, positive self-management of the LTC improved feelings of control and self-efficacy and in so doing improved depressed mood. Depressed patients have previously been reported to benefit most from self-efficacy gains

from LTC self-management training[31]; the importance of this construct for therapeutic purposes within the modern care setting of depression and LTC patients requires future investigation.

Our findings have a number of important implications for care models such as the National Health Services' (NHS') IAPT-LTC programme.[32] Our findings suggest that any care model of depression in LTC needs to take into account four overarching conclusions based on our data: (1) not all participants with depression and LTC view their mental and physical health as interconnected; therefore, there should be allowances in care plans for separate treatment (this is a view endorsed by Knowles and colleagues)[10]; (2) key features of depression that affect LTC management and need to be considered when planning treatment and referring to psychological services are social withdrawal and lack of motivation to self-manage or access healthcare; (3) key features of LTCs that worsen depression that should be discussed during care planning are pain, the unpredictability of future health and progressive disability; (4) positive self-management of LTC could improve feelings of control and self-efficacy and therefore mood, and should be encouraged.

A strength of our study was in the recruitment of participants from GP practices ensuring all our participants had received clinical diagnoses of their health conditions. To understand the differences in depression across different LTCs we excluded participants with more than one LTC of interest and with a serious or life-limiting diagnosis such as cancer, stroke or dementia. Due to the increased risk of physical illness with age, we recruited participants within an age bracket of 55–75 years; this enabled us to focus on the experiences of adults in mid to late life. The study was confined to North and North West London; however, we recruited from across contrasting areas to elicit a range of experiences. Despite our best efforts, unfortunately our arthritis sample was predominately female (90%); however, this may partly reflect the greater incidence of osteoarthritis in women in the UK.[33] Our depression and CHD group was small (n=5); therefore, our findings may not reflect the diversity of views that a larger sample might have attained. Also, as part of our reflexive practice, we noted that as researchers our experience and identities as female academic researchers working in London, of White (HR, RF, GB) and mixed (LP) ethnic background, is likely to have shaped our understanding, prioritisation and interpretation of the data. The trustworthiness of our analysis, however, has been enhanced through researcher triangulation, drawing on the multiple expertise of the research team. Moreover, we used multiple levels to the analysis process from use of the reflective diary, through to the generation of themes, and the comparative exercise through the charting of codes. Participant checking was not undertaken which could have added a useful check to our interpretive work.

In conclusion, the results of our face-to-face interview study of depression in those with and without a physical LTC revealed that overall, key diagnostic symptoms of

depression (anhedonia, sadness, etc) were experienced by all our participants regardless of LTC. However, the LTC did interact with depression by compounding somatic, cognitive and emotional symptoms, increasing disability and reducing independence, and hindering attempts at coping with mental illness. Our findings have implications for the development and fine-tuning of mental health and LTC care programmes.

**Contributors** LP was responsible for funding acquisition and conceived the study design and acts as the guarantor for the study. GB and RF advised on the study methods. LP and HR collected the data. LP and RF led the data analysis. GB and HR contributed to the data analysis. LP wrote the initial manuscript draft. RF, GB and HR edited the manuscript.

**Funding** LP and HR were funded by the UK Economic and Social Research Council (ES/N001478/1) for this project.

**Competing interests** None declared.

**Patient consent for publication** Not applicable.

**Ethics approval** This study involves human participants and was approved by NHS London–Fulham Research Ethics Committee (ID: 18/LO/2150). Participants gave informed consent to participate in the study before taking part.

**Provenance and peer review** Not commissioned; externally peer reviewed.

**Data availability statement** Data are available in a public, open access repository. Data for this study are archived with the UK Data Service.

**ORCID iDs**
Lydia Poole http://orcid.org/0000-0003-1325-7107
Rachael Frost http://orcid.org/0000-0003-3523-0052
Georgia Black http://orcid.org/0000-0003-2676-5071

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
