## [Reviewer comments · BMJ Open]

ARTICLE DETAILS

TITLE (PROVISIONAL)	The experience of depression in older adults with and without a physical long-term condition: findings from a qualitative interview study.
AUTHORS	Poole, Lydia; Frost, Rachael; Rowlands, Hannah; Black, Georgia

VERSION 1 – REVIEW

REVIEWER	Hale, Leigh University of Otago, School of Physiotherapy
REVIEW RETURNED	08-Dec-2021

GENERAL COMMENTS	Thank you for the opportunity to read this interesting and well written manuscript. It concludes with clearly articulated suggestions for future practice that are worthy of consideration by health care professionals. I have only a few minor recommendations to consider for revision: (1) a COREQ (COnsolidated criteria for REporting Qualitative research) Checklist should be completed, and items currently missing from this checklist addressed(2) A section on reflexivity of the research team should be included (as who does the data collection and analysis does influence these processes)(3) To me, the "participant selection and recruitment section" was confusing as the inclusion / exclusion criteria were not explicitly stated, the section leads with the outcome of the recruitment strategy (and this should rather be stated in the beginning of the results section) before it described the recruitment strategies. How participants were purposively sampled is not clearly explained.(4) What steps were taken to establish the trustworthiness of the analysis
---

REVIEWER	Eakin, Michelle Johns Hopkins University Bloomberg School of Public Health, Health, Behavior and Society
REVIEW RETURNED	20-Dec-2021

GENERAL COMMENTS	This is an interesting qualitative exploration of the link between depression and different chronic diseases. The results are intriguing and describe the individual challenges of managing comorbid depression and chronic illness. There are some questions about methodology outlined below. 1. Overall it is recommended that the authors follow some qualitative reporting guidelines for their manuscript, such as the CORE-Q. There are a number of issues missing. Below is a brief list but it is recommended that the author do a complete review.
--

	2. Details about the training and experience with qualitative research are needed for the interviewers. In addition it would be helpful to clarify if there was any relationship between the interviewer and participants. 3. More information about the coding process is needed. It is unclear if a code book was developed. How was triangulation assessed? Were there multiple coders? Who did the coding? How were discrepancies handled. 4. My main concern is how data from the vignettes were analyzed compared to the open ended questions? What information in the results came from the vignettes? How were they integrated into the analyses. 5. The lack of information about the data collected from the vignettes is confusing and it is hard to see how they guided the results. 6. Reasons for ineligibility should be included since 137 were interested but only 45 were eligible. 7. Figure 1 did not appear- I just see a caption with no figure
--	---

VERSION 1 – AUTHOR RESPONSE

Reviewer: 1 Prof. Leigh Hale, University of Otago

Comments to the Author:

Thank you for the opportunity to read this interesting and well written manuscript. It concludes with clearly articulated suggestions for future practice that are worthy of consideration by health care professionals.

Thank you for taking the time to review our manuscript. We appreciate your positive feedback.

6. I have only a few minor recommendations to consider for revision: a COREQ (COnsolidated criteria for REporting Qualitative research) Checklist should be completed, and items currently missing from this checklist addressed

Thank you for this advice. I have now completed this checklist and have included as a supplementary file.

7. A section on reflexivity of the research team should be included (as who does the data collection and analysis does influence these processes)

A reflective diary was kept, this is now clarified on p.7 of the methods, and described in relation to the analysis on p.8. In addition, we have added a description of reflexivity to p.24 of the discussion.

8. To me, the "participant selection and recruitment section" was confusing as the inclusion / exclusion criteria were not explicitly stated, the section leads with the outcome of the recruitment strategy (and this should rather be stated in the beginning of the results section) before it described the recruitment strategies. How participants were purposively sampled is not clearly explained.

We are sorry this was confusing to follow. We have removed the sample size from the participant selection and recruitment section and have moved this to the results (p.9). We have also clarified our purposive sampling procedure and moved it to the end of this section (p.6). We have used rephrased the inclusion/exclusion criteria to make these easier to locate (p.5/6).

9. What steps were taken to establish the trustworthiness of the analysis

Thank you for this comment. We have now discussed researcher triangulation and the multiple levels to our analytic process in the discussion on p.24.

Reviewer: 2

Dr. Michelle Eakin, Johns Hopkins University Bloomberg School of Public Health

Comments to the Author:

This is an interesting qualitative exploration of the link between depression and different chronic diseases. The results are intriguing and describe the individual challenges of managing comorbid depression and chronic illness. There are some questions about methodology outlined below.

Thank you for taking the time to review our manuscript. We appreciate your feedback and hope to have addressed your concerns adequately below.

10. Overall, it is recommended that the authors follow some qualitative reporting guidelines for their manuscript, such as the CORE-Q. There are a number of issues missing. Below is a brief list but it is recommended that the author do a complete review.

Thank you for this advice. I have now completed this checklist and have included as a supplementary file.

11. Details about the training and experience with qualitative research are needed for the interviewers. In addition, it would be helpful to clarify if there was any relationship between the interviewer and participants.

Thank you for this suggestion. This has now been clarified on p.7 of the methods.

12. More information about the coding process is needed. It is unclear if a code book was developed. How was triangulation assessed? Were there multiple coders? Who did the coding? How were discrepancies handled.

Thank you for this comment. We have clarified the coding process on p.8/9 of the methods. In line with reviewer 1's comment 9, we have included a discussion of triangulation on p.24.

13. My main concern is how data from the vignettes were analyzed compared to the open ended questions? What information in the results came from the vignettes? How were they integrated into the analyses.

The discussion surrounding the vignettes was included in the transcripts. In practice, the vignettes worked as an ice-breaker, allowing participants a way to initiate the conversation about their own mental health. Some participants did not engage well with the vignettes and therefore not much was said before the semi-structured questions began. For others, it sparked a series of memories which allowed them to recall earlier life events and experiences. During the analysis there was no separation or division of the data; each transcript was handled as a complete data piece. We have now clarified the vignette data on p.8.

14. The lack of information about the data collected from the vignettes is confusing and it is hard to see how they guided the results.

We hope to have addressed this in response to your comment 13 above.

15. Reasons for ineligibility should be included since 137 were interested but only 45 were eligible.

Reasons for ineligibility are now described on p.9 of the results.

16. Figure 1 did not appear- I just see a caption with no figure

Apologies for this. We have uploaded the Figure as a .jpg file in line with the Editor's request (comment 4 above).

VERSION 2 – REVIEW

REVIEWER	Hale, Leigh University of Otago, School of Physiotherapy
REVIEW RETURNED	27-Jan-2022

GENERAL COMMENTS	The revision has strengthened this paper and all my recommendations were addressed. Just a few very minor suggested changes to make: Page 7: Suggest change “The interviewer kept a reflective diary” to “The primary interviewer (HR) kept a reflective diary”. Page 8: Considering your data analysis method is thematic analysis recommend change “conceived as emerging from the” to “conceived as developing from the” Page 8: Change ‘As such, this data was not separated” to “As such, these data were not separated”
--